# REGLO: Provable Neural Network Repair for Global Robustness Properties

**Feisi Fu**[1]*, **Zhilu Wang**[2]*, **Jiameng Fan**[1], **Yixuan Wang**[2],
**Chao Huang**[3], **Qi Zhu**[2], **Xin Chen**[4], **Wenchao Li**[1]

[1]Department of Electrical and Computer Engineering, Boston University, Boston, MA, USA
Emails: {fufeisi, jmfan, wenchao}@bu.edu
[2]Department of Electrical and Computer Engineering, Northwestern University, Evanston, IL, USA
Emails: {zhilu.wang, yixuanwang2024}@u.northwestern.edu, qzhu@northwestern.edu
[3]Department of Computer Science, University of Liverpool, Liverpool, UK
Email: chao.huang2@liverpool.ac.uk
[4] University of Dayton, Dayton, OH, USA
Email: xchen4@udayton.edu

## Abstract

We present REGLO, a novel methodology for repairing neural networks to satisfy *global robustness* properties. In contrast to existing works that focus on local robustness, i.e., robustness of individual inputs, REGLO tackles global robustness, a strictly stronger notion that requires robustness for all inputs within a region. Leveraging an observation that any counterexample to a global robustness property must exhibit a corresponding large gradient, REGLO first identifies violating regions where the counterexamples reside, then uses verified robustness bounds on these regions to formulate a robust optimization problem to compute a minimal weight change in the network that will provably repair the violations. Experimental results demonstrate the effectiveness of REGLO across a set of benchmarks.

## 1 Introduction

Motivated by the fragility of deep neural networks (DNNs) to small input perturbations known as *adversarial examples* [1], there is a large, growing body of research on measuring, verifying, and improving the robustness of DNNs against those perturbations [2, 3, 4, 5, 6, 7, 8, 9, 10]. Various notions of robustness have been considered [11, 12, 13, 14, 15], and can be largely categorized into two groups: *local robustness* and *global robustness*. Local robustness is about the robustness of individual input points. Intuitively, it means that for an input $x$, a small change of the input (e.g., any $p$-norm-bounded $\Delta x$) would not result in a significant change in the output (e.g., change of a classification result). On the other hand, global robustness stipulates robustness on *all* points within a given input region $X$. Global robustness is strictly stronger than local robustness, and has the advantage of enforcing robustness on unseen inputs within the given input region.

Techniques developed for verifying local robustness [16, 17, 18, 10] are often leveraged as a subroutine when verifying global robustness and probabilistic notion of robustness. For instance, Weng et al. [18] derives a probabilistic certificate based on the worst-case certificate computed for local robustness. A standard way to compute global robustness bounds is to first construct a twin-network, i.e., two copies of the original network side by side, where one input represents $x$ and the other input represents its adversarial perturbation $\Delta x$, and then apply techniques for computing local robustness

---

*The first two authors contributed equally to this paper.

2022 Trustworthy and Socially Responsible Machine Learning (TSRML 2022) co-located with NeurIPS 2022.

bounds to the twin-network for any $x \in X$ and $\Delta x$ within some given perturbation set [14, 19]. Global robustness is known to be harder to verify than local robustness, due to encoding two copies of the network and a larger input domain [14]. Recently, Wang et al. [20, 21] improved the efficiency of global robustness verification by exploiting the interleaving dependencies in the twin-network encoding. For ReLU DNNs, it is also possible to exploit their piecewise linearity and enumerate the activation patterns to search for counterexamples to a global robustness specification [22].

Recognizing the importance of global robustness, various methods have also been proposed to train networks with improved global robustness. Leino et al. [23] present a method for training and constructing globally-robust classification networks with an additional output class that labels inputs as "non-locally-robust". Chen et al. [13] use a counterexample-guided framework to train classifiers that satisfy global robustness properties. In addition, the notion of *individual fairness* (IF), which requires two inputs that differ only on some sensitive features to have similar outputs, can be viewed as a global robustness property [24]. Benussi et al. [25] presents an MILP formulation whose solution can be used to verify IF properties and guide the training process by modifying the training loss. Existing works also leverage adversarial-training schemes by using a discriminator to force the classifier to be unbiased towards the sensitive features [26, 27, 28]. In general, verification-in-the-loop training approaches can be prohibitively expensive given the high cost of global robustness verification. Moreover, training-based methods cannot guarantee the satisfaction of global robustness properties.

In this paper, we consider the problem of *repairing* a trained DNN to satisfy a given global robustness property. Repair and verification are two sides of the same coin – if a problem warrants (formal) verification, then any bug discovered during verification should necessitate fixing. Existing DNN repair methods mainly consist of weight modification [29, 30, 31], either via constraint solving or fine-tuning, and DNN architecture extension [32, 33, 34]. Compared with training-based approaches, repair can be applied only once as a post-hoc modification and does not require access to the training data. A repair method is considered *sound* if it can guarantee the removal of the discovered violations or the satisfaction of a given property.

**Our contributions:** We propose REGLO, the *first DNN repair technique with provable guarantees on satisfying global robustness properties*. The key idea of REGLO is to leverage an observation that any counterexample of a global robustness property would have a large gradient that indicates the violation, and use verified robustness bounds on the corresponding violating region to formulate a robust optimization problem to compute a minimal weight change in the last hidden layer of the network to fix the violation. For piecewise linear DNNs, our approach is both sound and complete – the resulting network is guaranteed to satisfy the given global robustness property, and a repair is guaranteed to be found. We detail our approach below, starting with the technical preliminaries.

## 2 Background

### 2.1 Deep Neural Networks (DNNs)

An $R$-layer feed-forward DNN $f : \mathcal{X} \to \mathcal{Y}$ is a composition of linear functions and activation function $\sigma$, where $\mathcal{X} \subseteq \mathbb{R}^m$ is a bounded input domain and $\mathcal{Y} \subseteq \mathbb{R}^n$ is the output domain. The weights and biases of the linear function are parameters of the DNN. We call the first $R - 1$ layers hidden layers and the $R$-th layer the output layer. We use $z_j^i$ to denote the $i$-th neuron (before activation) in the $j$-th hidden layer.

For DNNs that use only the ReLU activation function $\sigma(x) = \max(x, 0)$, we call them ReLU DNNs. For any neuron $z_j^i$, we say the neuron is activated for an input if and only if the neuron's value $\sigma(z_j^i) = z_j^i$. We use a binary variable $\alpha_j^i$ to represent the activation status of $z_j^i$ (where $\alpha_j^i = 1$ means the neuron is activated). The set of activation statuses $\{\alpha_j^i\}$ of all the neurons is called an activation pattern. It is known that an $\mathbb{R}^m \to \mathbb{R}$ function is representable by a ReLU DNN *if and only if* it is a continuous piecewise linear (CPWL) function [35].

### 2.2 Linear Regions

A *linear region* is the set of inputs that are subject to the same activation pattern in a ReLU DNN [36].

**Lemma 1.** *[37] Consider a ReLU DNN $f$ and an input $x \in \mathbb{R}^m$. For every neuron $z_j^i$, it induces a feasible set*

$$\mathcal{A}_j^i(x) = \begin{cases} \{\bar{x} \in \mathbb{R}^m | (\nabla_x z_j^i)^T \bar{x} + z_j^i \\ -(\nabla_x z_j^i)^T x \geq 0\} & \text{if } z_j^i \geq 0 \text{ or } \alpha_j^i = 1 \\ \{\bar{x} \in \mathbb{R}^m | (\nabla_x z_j^i)^T \bar{x} + z_j^i \\ -(\nabla_x z_j^i)^T x \leq 0\} & \text{if } z_j^i < 0 \text{ or } \alpha_j^i = 0 \end{cases} \tag{1}$$

*The intersection $\mathcal{A}(x) = \bigcap_{i,j} \mathcal{A}_j^i(x)$ is the linear region that includes $x$. Note that $\mathcal{A}(x)$ is essentially the H-representation of the corresponding convex polytope.*

## 2.3 Global Robustness Property

We consider a global robustness property $\mathbf{P}$ on $X \subseteq \mathcal{X}$.

**Definition 1** (Global Robustness). *A DNN $f$ satisfies a global robustness property $\mathbf{P}(X, \Omega, \epsilon)$ on an input set $X \subseteq \mathcal{X}$ along with a perturbation set $\Omega \subseteq \mathbb{R}^m$ if and only if for any $x \in X$ and $\Delta x \in \Omega$, $\|f(x) - f(x + \Delta x)\| \leq \epsilon$ holds.*

**Definition 2** (Norm-Bounded Global Robustness). *A DNN $f$ is $(\delta, \epsilon)$-globally robust on an input set $X \subseteq \mathcal{X}$ if and only if for any $x \in X$, we have that $\|\Delta x\| \leq \delta \Rightarrow \|f(x) - f(x + \Delta x)\| \leq \epsilon$.* [1]

This is an instantiation of the global robustness property with $\Omega = \{\Delta x \mid \|\Delta x\| \leq \delta\}$. The notion of *individual fairness* [38, 24] can also be viewed as a global robustness property, by taking $\Omega = \{\Delta x \mid \Delta x_{\mathrm{NF}} = 0\}$ where SF indicates sensitive features and NF indicates the remaining, non-sensitive features, as follows.

**Definition 3** (Individual Fairness). *A DNN $f$ is $\epsilon$-fair with respect to some sensitive input features SF if and only if for any $x \in \mathcal{X}$, if $x_{SF} = (x + \Delta x)_{NF}$, then $\|f(x) - f(x + \Delta x)\| \leq \epsilon$.*

## 2.4 Neural Network Repair

We are now ready to define the repair problem.

**Definition 4** (Repair for Global Robustness Property). *Given a global robustness property $\mathbf{P}(X, \Omega, \epsilon)$ and a target DNN $f \not\models \mathbf{P}(X, \Omega, \epsilon)$, the repair problem is to find a modified DNN $\widehat{f}$ such that $\widehat{f} \models \mathbf{P}(X, \Omega, \epsilon)$.* [2]

## 2.5 Verification for Global Robustness Property

A standard way to verify global robustness is to reduce it to verifying local robustness by constructing a twin-network [39]. *ITNE* [20, 21] is the state-of-the-art verification technique that uses an interleaving twin-network encoding approach where two copies of the neural network are encoded side-by-side with extra interleaving dependencies added between them. Bound propagation techniques [17, 40, 10, 41] can then be applied to compute the output bound for a given input area on the twin-network. In REGLO, we use global robustness verification to guide the repair process which we will describe in detail in the next section.

## 3 The REGLO Approach

We present REGLO's approach below, starting with an observation that allows us to identify repair regions that violate a given global robustness property. A preview of the high-level approach in REGLO is also given in Figure 1.

## 3.1 Key Observation

A counterexample to a global robustness property $\mathbf{P}(X, \Omega, \epsilon)$ is a pair of inputs $(x, \Delta x)$ that violates the property $\mathbf{P}$, i.e., $x \in X$ and $\Delta x \in \Omega$ but $\|f(x) - f(x + \Delta x)\| \geq \epsilon$. Based on the following

---

[1]The norm in this definition can be on any $p$ norm.

[2]We consider the case where $f$ and $\widehat{f}$ share the same structure, i.e. the same number of layers and the same number of neurons.

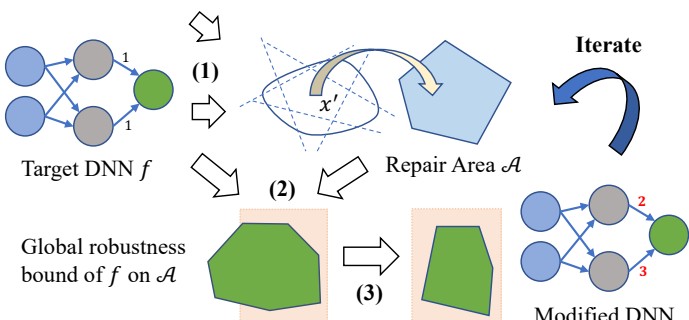

Figure 1: An illustration of REGLO's verification-guided algorithm: in each iteration, (1) identify repair areas that violate the global robustness property $\mathbf{P}$, (2) compute the global robustness bound for each repair area (in green with the light coral area showing the desired bound according to $\mathbf{P}$), and (3) solve a convex robust optimization problem to modify the last-layer weights of the target DNN so that the modified DNN is guaranteed to satisfy the global robustness property on those repair areas.

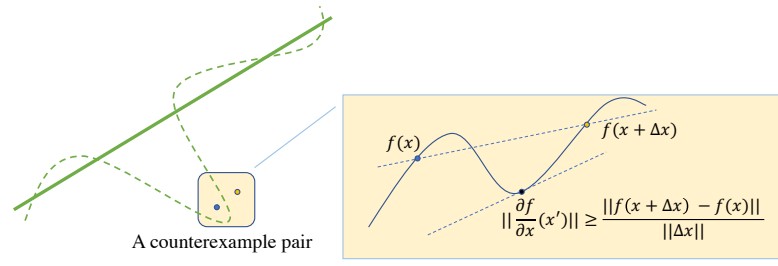

Any counterexamples indicate points with a large gradient.

Figure 2: The solid green line and the dash green line is the decision boundary for data distribution and the DNN's decision boundary respectively. The deviation of DNN's decision boundary leads to a violation of the global robustness property. Our key observation is that if $(x, x + \Delta x)$ is a counterexample pair of a global robustness property for a DNN $f$, then there exists $x'$ with a large gradient.

*Mean Value Inequality Theorem* [42], we can convert the search of a potential violation area (an area that contains a counterexample pair) for a global robustness property to the search of a single point with a large gradient, as illustrated in Figure 2.

**Theorem 1** (Mean Value Inequality [42]). *For a continues function $f : [a, b] \to \mathbb{R}^n$, if $f$ is differentiable on (a, b), then*

$$\|f(b) - f(a)\| \leq (b - a) \sup_{x \in (a,b)} \|f'(x)\| \tag{2}$$

Directly applying the *Mean Value Inequality Theorem* to a global robustness property on a DNN, we obtain the following corollary.

**Corollary 1** (Gradient Features of Global Robustness Properties). *For a DNN $f$ and a global robustness property $\mathbf{P}(X, \Omega, \epsilon)$, if there is a counterexample $(x, \Delta x)$ such that $\Delta x \in \Omega$ and $\|f(x + \Delta x) - f(x)\| \geq \epsilon$, then there exists a differentiable point $x'$ between $x$ and $x + \Delta x$, such that $\|x - x'\| \leq \frac{dia(\Omega)}{2}$ and $\|\frac{\partial f}{\partial x}(x')\| > \frac{\epsilon}{dia(\Omega)}$, where $dia(\Omega)$ is the diameter of $\Omega$.* [3]

**Remark**: Note that this corollary specifies a necessary but not sufficient condition. In other words, the presence of a counterexample must exhibit a large corresponding gradient but the reverse is not necessarily true.

---

[3]Here we take the operator norm of a matrix.

The *Mean Value Inequality Theorem* requires differentiability of the function $f$ on $(a, b)$. For non-differentiable DNNs, the non-differentiability comes from its activation functions, e.g., ReLU, LeakReLU, Heaviside, or Maxpooling. Since the measure of the non-differentiable area is zero for those operations, we can still apply the *Mean Value Inequality Theorem* in a piecewise manner to obtain the same result.

**Remark**: For individual fairness, we can define the norm $\|x\| = \|x\|_F$ if $\|x\|_{NF} = 0$ and else $\|x\| = +\infty$, and only consider the gradient with respect to the sensitive features $\|\frac{\partial f}{\partial x}(x')\| = \|\frac{\partial f}{\partial x_{SF}}(x')\|$.

## 3.2 Repair Areas

For a ReLU DNN and the $L_\infty$ output bound, we can encode the search of the maximal gradient as a mixed-integer programming (MILP) problem [43, 44]:

$$\max c(x, \Delta x) = \frac{1}{\nu}\|z_R - z'_R\|_\infty \tag{3}$$

$$\begin{cases} z_{j+1} \geq 0, z_{j+1} \leq B\alpha_{j+1} \\ z_{j+1} \geq \theta_j z_j + b_j \\ z_{j+1} \leq \theta_j z_j + b_j + B(1 - \alpha_{j+1}) \\ z'_{j+1} \geq 0, z'_{j+1} \leq B\alpha_{j+1} \\ z'_{j+1} \geq \theta_j z'_j + b_j \\ z'_{j+1} \leq \theta_j z'_j + b_j + B(1 - \alpha_{j+1}) \\ \alpha_j = \{\alpha_j^i\}, \alpha_j^i \in \{0, 1\} \end{cases}$$

for $0 \leq j \leq R - 1$,

$$x \in X, -\nu \leq \|\Delta x\|_\infty \leq \nu, z_0 = x, z'_0 = x + \Delta x,$$

where $B$ is a big enough number and $\nu$ is a small enough number. [4] By solving the MILP problem (3), we obtain the optimal input $x^*$, the optimal activation pattern $\alpha^*$, and the largest gradient norm $c^*$.

Our goal is to find all the linear regions with gradients larger[5] than $\frac{\epsilon}{dia(\Omega)}$. We store the linear region $\{x \mid Ax \leq b\}$ that contains $x^*$ which can be obtained from Lemma 1 for $c^* > \frac{\epsilon}{dia(\Omega)}$. Then we add the following additional constraint to the MILP problem (3), which is used to exclude the same activation pattern $\hat{\alpha}$ that we have already found, and solve the MILP problem with the additional constraint to find a new violating linear region.

$$\sum_{i,j}[\hat{\alpha}_{i,j}\alpha_{i,j} + (1 - \hat{\alpha}_{i,j})(1 - \alpha_{i,j}) - 1] \leq -1 \tag{4}$$

Since the number of linear regions on $X$ is finite, we can find all the linear regions with gradient greater than $\frac{\epsilon}{dia(\Omega)}$ by iteratively solving MILPs and accumulating the exclusion constraint (4) until the optimal solution $c^*$ is smaller than $\frac{\epsilon}{dia(\Omega)}$. Note that while the total number of linear regions can be very large, the number of violating regions is typically much smaller.

For non-CPWL DNNs, we can use random sampling to search for input $x'$ that satisfies the violation constraint $\|\frac{\partial f}{\partial x}(x')\| > \frac{\epsilon}{dia(\Omega)}$. If the target DNN $f$ is twice differentiable, we can apply a projected gradient descent method to improve the sample efficiency. For CPWL DNNs, we can also start with random sampling and then pivot to the more expensive MILP-based method for better search efficiency.

Now we can define the repair area $\mathcal{A}$ as follows. $\mathcal{A} = \{x \mid Ax \leq b\}$ by solving MILP (3) or $\mathcal{A} = \{x \mid \|x - x'\| \leq \frac{\delta}{2}\}$ via random sampling. We use $\{\mathcal{A}_i\}_{i \in I}$ to denote all the repair areas found via the aforementioned procedure.

---

[4]$B$ is used for encoding a ReLU neural network as an MILP and $\nu$ is for computing the gradient.

[5]Note that the gradients of all the points inside a linear region are the same for a ReLU DNN.

### 3.3 Verification-Guided Constraints

For each repair area $\mathcal{A}_i$, we apply *ITNE* [21] to estimate $\epsilon_i^*$, which is an over-approximation of the global robustness bound on $\mathcal{A}_i$. $\epsilon_i^*$ is the optimal value of the following optimization problem.

$$\begin{cases} \max_{[\Delta x, \Delta z_i] \in \Omega \times Z_i} \|(\theta + \Delta\theta)\Delta z_i\| \\ Z_i = \{\Delta z \mid D_i^l \Delta x + e_i^l \leq \Delta z \leq D_i^u \Delta x + e_i^u\} \end{cases} \tag{5}$$

where $\theta$ is the weight in the last hidden layer of the DNN, $D_i^l, e_i^l, D_i^u$ and $e_i^u$ are parameters for the linear bounds of $\Delta z$, and $\Delta z$ is the neurons value difference of the last hidden layer between $x$ and $x + \Delta x$.

**Remark:** We consider weight modification in the last hidden layer because it does not change the activation pattern of any input and in turn preserves the boundaries of the linear regions. Thus, by repairing the violating linear regions iteratively, we can guarantee satisfaction of the given global robustness property. In theory, we can also consider modifying the weights of an intermediate layer. However, the objective function of optimization problem (5) will include the subsequent layers and the optimization problem will no longer be convex.

### 3.4 Repair as Robust Optimization

We use $\Delta\theta$ to denote the weight change to the DNN's last hidden layer. In order to preserve the functionality (e.g., accuracy) of the network, we aim to find a *minimal* weight change $\Delta\theta$ that can guarantee the satisfaction of global robustness property on all the repair areas $\{\mathcal{A}_i\}_{i \in I}$. Formally, $\Delta\theta$ is the solution to the following optimization problem.

$$\begin{cases} \min \|\Delta\theta\| \\ \max_{[\Delta x, \Delta z_i] \in \Omega \times Z_i} \|(\theta + \Delta\theta)\Delta z_i\| \leq \epsilon \\ \text{where } Z_i = \{\Delta z \mid D_i^l \Delta x + e_i^l \leq \Delta z \leq D_i^u \Delta x + e_i^u\} \end{cases} \tag{6}$$

**Property 1.** *Optimization problem (6) is convex and thus any local minimum also achieves the global minimum.*

This minimization problem with inner maximal constraints is a form of *robust optimization* [45]. For minimizing the $L_1$ or $L_\infty$ norm of $\Delta\theta$, such *robust optimization* problems can be solved by taking the *robust counterpart* of the inner constraints and converting it to a linear programming (LP) problem.

### 3.5 Repair via Barrier Method

For general $L_p$ norm of $\Delta\theta$, we apply the *barrier method* from [46] and formulate it as an unconstrained convex optimization problem. We use $[\Delta x_i^*(\Delta\theta), \Delta z_i^*(\Delta\theta)]$ and $\epsilon_i^*(\theta + \Delta\theta)$ to denote the optimal solution and the optimal value respectively for optimization problem (5). We have, for a sufficiently large $t$, the solution of the following barrier problem converges to the solution of the optimization problem (6).

$$\min_{\Delta\theta} \|\Delta\theta\| - \frac{1}{t} \sum_{i \in I} \log(\epsilon - \epsilon_i^*(\theta + \Delta\theta)) \tag{7}$$

To solve the optimization problem (7), we compute the gradient of $\epsilon_i^*(\theta + \Delta\theta)$ by

$$\frac{\partial \epsilon_i^*(\theta + \Delta\theta)}{\partial \Delta\theta} = \frac{\partial \|(\theta + \Delta\theta)^T \Delta z_i\|}{\partial \Delta\theta}\Big|_{\Delta z_i = \Delta z_i^*(\Delta\theta)} \tag{8}$$

### 3.6 Iterative Repair

In the previous sections, we enumerate all the repair areas $\{\mathcal{A}_i\}_{i \in I}$ that violate $\mathbf{P}(X, \Omega, \epsilon)$ and repair these areas via weight modification. However, it is possible that the repair increases the gradient $\|\frac{\partial f}{\partial x}(x')\|$ for $x' \in X \setminus \cup_{i \in I} \mathcal{A}_i$ and results in a new violation to $\mathbf{P}(X, \Omega, \epsilon)$. To ensure a sound repair, we iteratively search for repair areas to repair. When solving the optimization problem, we consider both the previously repaired areas and the new repair areas.

We call one iteration, including the search of repair areas and the repair itself, a *Single Iteration Repair*. The *Iterative Repair* algorithm is given in Algorithm 1.

---

**Algorithm 1** Iterative Repair for ReLU DNN

---

**Input**: The target ReLU DNN $f$ and the global robustness property $\mathbf{P}(X, \Omega, \epsilon)$.
**Parameter**: Maximum iteration $T$ and maximum number of repair areas $M$.
**Output**: The repaired DNN.
 1: Let $t = 0$.
 2: Let repair areas $I = \emptyset$
 3: **while** $t < T$ **do**
 4:   **while** $|I| < M$ **do**
 5:     Let $c^*$ and $x^*$ be the optimal value and solution of MILP (3), respectively.
 6:     **if** $c^* \leq \frac{\epsilon}{dia(\Omega)}$ **then**
 7:       Break
 8:     **end if**
 9:     Let $\mathcal{A}_i$ be the linear region containing $x^*$ computed using Lemma 1.
10:     $I$.add($\mathcal{A}_i$).
11:     Add a new exclusion constraint (4) to MILP (3).
12:   **end while**
13:   Formulate the constraints on all repair areas $I$ as a robust optimization (6), solve it via Algorithm 2 (in the Appendix), and obtain an optimal solution $\Delta\theta$.
14:   Update $f$'s last-layer weight $\theta = \theta + \Delta\theta$.
15: **end while**
16: **return** $f$ with updated last-layer weight.

---

# 4 Theoretical Guarantees

In this section, we present the theoretical guarantees that `REGLO` provides, and point the readers to proofs of the theorems in the Appendix.

## 4.1 Completeness Guarantees

**Theorem 2** (Completeness Guarantees)**.** *We have the following completeness guarantees for* `REGLO`*:*

 1. *For a Single Iteration Repair,* `REGLO` *can always find a solution to optimization problem (6).*

 2. *For an Iterative Repair on a piecewise linear DNN,* `REGLO` *always terminates with no more repair areas to be found by solving MILP (3).*

## 4.2 Soundness Guarantees

Ideally, if the target DNN is a ReLU DNN, we can enumerate all the linear regions that violate the global robustness property $\mathbf{P}(X, \Omega, \epsilon)$ by solving multiple MILPs and applying `REGLO`. By *Iterative Repair*, `REGLO` will terminate when no more repair area can be found. Thus, we have the soundness guarantee for the resulting DNN.

Specifically, for a *Single Iteration Repair*, the weight change $\Delta\theta$ ensures the satisfaction of $\mathbf{P}(X, \Omega, \epsilon)$ on all the repair areas $\{\mathcal{A}_i\}_{i \in I}$. As we discussed in the *Iterative Repair Section*, the repair may increase the gradient $\|\frac{\partial f}{\partial x}(x')\|$ for $x' \in X \setminus \cup_{i \in I} \mathcal{A}_i$ and causes it to violate $\mathbf{P}(X, \Omega, \epsilon)$. We call any such violation in $X \setminus \cup_{i \in I} \mathcal{A}_i$ a *side effect* of our repair. The following theorem shows that we have guarantees on limiting the side effects of a *Single Iteration Repair*.

**Theorem 3** (Limited Side Effect for Single Iteration Repair)**.** *Given a global robustness property* $\mathbf{P}(X, \Omega, \epsilon)$*, a target DNN $f$, and weight change $\Delta\theta$ from a Single Iteration Repair, we have*

 1. *for any area $\mathcal{B} \subset \cup_{i \in I} \mathcal{A}_i \subset X$, $\widehat{f} \models \mathbf{P}(\mathcal{B}, \Omega, \epsilon)$;*

 2. *for any area $\mathcal{C} \subset X$ which is not a subset of $\cup_{i \in I} \mathcal{A}_i$, $\widehat{f} \models \mathbf{P}(\mathcal{C}, \Omega, \epsilon + 2L\|\Delta\theta\|\|X\|)$, where $L$ is the Lipschitz constant of $f$ from the input layer to the last hidden layer.*

**Corollary 2** (Soundness Guarantees for Repairing CPWL DNNs)**.** *Given a global robustness property* $\mathbf{P}(X, \Omega, \epsilon)$*, a piecewise linear DNN $f$, and weight change from Iterative Repair,* `REGLO` *will terminate with no more repair areas to be found and the resulting DNN $\widehat{f} \models \mathbf{P}(X, \Omega, \epsilon)$.*

For DNNs that are not piecewise linear, we have the following weaker soundness guarantee.

**Corollary 3** (Soundness Guarantee for Repairing General DNNs). *Given a global robustness property* $\mathbf{P}(X, \Omega, \epsilon)$, *a DNN* $f$, *and weight change* $\Delta\theta$ *from Iterative Repair,* `REGLO` *returns a DNN* $\widehat{f} \models$ $\mathbf{P}(\mathcal{C}, \Omega, \epsilon + 2L\|\Delta\theta\|\|X\|)$ *for any repair area* $\mathcal{C}$.

## 5 Experiments

Our prototype tool is implemented in Python. We use Gurobi [47] to solve MILP (3) and use CVXPY [48] to solve optimization problem (5). The global robustness bounds used in (5) are derived by ITNE [21] with a bound propagation technique similar to $\beta$-CROWN [41] (without bound refinements for efficiency). The verification bounds (VBs) in the experiments are evaluated using ITNE with bound refinements for tighter estimations.

### 5.1 Baseline Methods

To the best of our knowledge, `REGLO` is the *first* study on DNN repair to satisfy a global robustness property. We consider the following five baseline methods for comparison with `REGLO`.

- ST: standard training.

- AT: adversarial training using PGD [3] on the training data.

- AT-G: adversarial training with counterexamples that violate the global robustness property (AT-G). The counterexamples are generated by applying PGD on randomly sampled points in $X$.

- ST+AT-G: standard training followed by an adversarial fine-tuning for the global robustness property. Fine-tuning is to fine-tune (train with a smaller learning rate) a pre-trained DNN with additional counterexamples generated similarly to those in AT-G.

- AT+AT-G: adversarial training followed by an adversarial fine-tuning for the global robustness property.

For `REGLO`, we consider two settings where the repair is applied after standard training and after adversarial training respectively.

- ST+REGLO: standard training followed by `REGLO`.

- AT+REGLO: adversarial training followed by `REGLO`.

### 5.2 Evaluation Metrics

Given that there is no efficient method for *exact* verification of a global robustness property, we use *ITNE* [20] to compute an upper-bound of the true global robustness $\epsilon^*$. In addition, we use PGD to evaluate the empirical robustness of randomly sampled inputs in $X$, as a lower-bound of $\epsilon^*$. We also report accuracy on testing data and runtime.

- VB: the verification bound given by *ITNE* for the global robustness property.

- PGD-B: the maximum norm difference on outputs between input pairs $(x, x+\Delta x)$ computed by PGD;

- PGD-R: the ratio of input pairs computed by PGD that violate the global robustness property.

- ERR: the mean absolute error on prediction for testing data (for regression problems);

- ACC: the accuracy on testing data (for classification problems).

- T(s): runtime in seconds.

**Remark:** Note that PGD-B $\leq \epsilon^* \leq$ VB, where $\epsilon^*$ is the unknown, true global robustness of the target DNN for $X$ and $\Omega$.

## 5.3 Benchmark Evaluations

We perform three benchmark evaluations, including individual fairness in a classification problem, as well as norm-bounded global robustness in a regression problem and a classification problem. All experiments were run on machines with CPUs similar to ten-core 2.6 GHz Intel Xeon E5-2660v3 without GPU. Details of the experimental setup such as DNN architectures and training hyperparameters can be found in the Appendix.

**German Credit (classification): repair for individual fairness.** We train a ReLU DNN on the

| | | ST | AT | AT-G | ST+AT-G | AT+AT-G | ST+REGLO | AT+REGLO |
|---|---|---|---|---|---|---|---|---|
| | VB | 12.5 | 4.69 | 7.28 | 9.0 | 4.52 | 0.29 | **0.26** |
| | PGD-B | 1.31 | 0.135 | 0.35 | 0.7 | 0.11 | 0.028 | **0.008** |
| All age | PGD-R | 75.5% | 26.9% | 45.7% | 57.4% | 1.5% | **0.0%** | **0.0%** |
| | ACC | **76.6%** | 69% | 76% | 68.3% | 69% | 76% | 69% |
| | T(s) | 25.5 | 38.3 | 26.5 | 25.5+18.4 | 38.3+17.9 | 25.5+51.8 | 38.3+50.3 |
| | VB | 1.11 | 0.42 | 0.68 | 1.04 | 0.4 | **0.08** | 0.12 |
| | PGD-B | 0.15 | 0.015 | 0.068 | 0.058 | 0.008 | 0.009 | **0.006** |
| Age below 24 | PGD-R | 7.8% | 0.5% | 2.4% | 0.3% | **0.0%** | **0.0%** | **0.0%** |
| | ACC | **76.6%** | 69% | 75.7% | 68.3% | 69% | 76.3% | 69% |
| | T(s) | 25.5 | 38.3 | 31.2 | 25.5+18.1 | 38.3+17.6 | 25.5+71.3 | 38.3+80.1 |
| | VB | 6.46 | 2.43 | 4.89 | 6.04 | 2.34 | **0.039** | 0.045 |
| | PGD-B | 0.68 | 0.084 | 0.43 | 0.28 | 0.044 | 0.0053 | **0.0015** |
| Age from 25 to 54 | PGD-R | 58% | 3.3% | 45.8% | 39.1% | **0.0%** | **0%** | **0.0%** |
| | ACC | **76.6%** | 69% | 76.3% | 69.7% | 69% | 69.3% | 69% |
| | T(s) | 25.5 | 38.3 | 34.2 | 25.5+17.3 | 38.3+18.2 | 25.5+55.5 | 38.3+47.9 |
| | VB | 2 | 0.75 | 1.13 | 1.88 | 0.73 | **0.057** | 0.11 |
| | PGD-B | 0.24 | 0.0265 | 0.093 | 0.11 | 0.016 | 0.0065 | **0.004** |
| Age from 55 to 64 | PGD-R | 23.4% | **0%** | 6.5% | 5.2% | **0.0%** | **0.0%** | **0.0%** |
| | ACC | **76.6%** | 69% | 76.3% | 71.7% | 69% | 75.7% | 69% |
| | T(s) | 25.5 | 38.3 | 30.9 | 25.5+18.5 | 38.3+17.6 | 25.5+47.9 | 38.3+39.4 |
| | VB | 2.23 | 0.84 | 1.19 | 2.1 | 0.81 | **0.065** | 0.094 |
| | PGD-B | 0.33 | 0.029 | 0.147 | 0.144 | 0.0192 | 0.0101 | **0.0033** |
| Age above 65 | PGD-R | 33.6% | **0.0%** | 15.8% | 7.4% | **0.0%** | **0.0%** | **0.0%** |
| | ACC | **76.6%** | 69% | 75.6% | 72.7% | 69% | 75.67% | 69% |
| | T(s) | 25.5 | 38.3 | 31.3 | 25.5+18.6 | 38.3+18.3 | 25.5+55.4 | 38.3+45.2 |

Table 1: Individual Fairness repair on German Credit for different age groups. It can be observed that after REGLO's repair, global robustness-related metrics including VB, PGD-B, and PGD-R are significantly reduced with little or no accuracy drop.

German Credit dataset [49] to predict the credit risks (good or bad) for a person based on input features. An ideal DNN predictor should be fair with respect to the sensitive input feature 'age', that is the resulting DNN should satisfy $\mathbf{P}(X, \Omega, \epsilon)$ for $\Omega = \{\Delta x \mid \Delta x_{SF} = 0\}$ with $\epsilon = 0.01$, where $SF$ are all input features other than 'age'. We consider the individual fairness properties on input domain $\mathcal{X}$ as well as regions based on age groups: $X_0 = \{x \mid x_{age} \leq 24\}$, $X_1 = \{x \mid 25 \leq x_{age} \leq 54\}$, $X_2 = \{x \mid 55 \leq x_{age} \leq 64\}$, or $X_3 = \{x \mid 65 \leq x_{age}\}$ according to [50]. For REGLO, we search for repair areas by random sampling on $X$ and choose 30 areas to repair. The results are shown in Table 1.

**Auto MPG (regression): sound repair for norm-bounded global robustness.** We train a ReLU DNN on the Auto MPG dataset [49] to predict the fuel efficiency (mile per gallon) of a car. We consider a norm-bounded global robustness property $\mathbf{P}(X, \Omega, \epsilon)$, where $\Omega = \{\Delta x \mid \|\Delta x\| \leq \delta\}$, $\epsilon = 1.5$, $\delta = 0.05$ and $X$ is the smallest hyper-rectangle that contains all the training inputs. For REGLO, we use MILP (3) to search for repair areas and apply an *Iterative Repair* (Algorithm 1). REGLO terminates after 2 iterations of repair and in total, 4 repair regions with different ReLU activation patterns were found during the repair. The results are shown in Table 2. [6]

**MNIST (classification): repair for norm-bounded global robustness.** We train a convolutional DNN with ReLU units on the MNIST dataset [51]. We consider a norm-bounded global robustness property $\mathbf{P}(X, \Omega, \epsilon)$, where $\Omega = \{\Delta x \mid \|\Delta x\| \leq \delta\}$, $\delta = 0.3$, and $\epsilon = 0.3$.

---

[6]Since Auto MPG is a dataset for regression and there is no adversarial example defined for such a dataset, adversarial training (AT) cannot be applied.

|        | ST    | AT-G  | ST+AT-G | ST+REGLO |
|--------|-------|-------|---------|----------|
| VB     | 2.52  | 2.44  | 2.42    | **1.48** |
| PGD-B  | 2.10  | 2.15  | 1.68    | **1.34** |
| PGD-R  | 0.75% | 0.85% | 0.40%   | **0.0%** |
| ERR    | 1.83  | 1.89  | **1.79**| 8.35     |
| T(s)   | 10    | 280   | 157     | 22       |

Table 2: Sound Repair on Auto MPG. Runtime includes both training and repair. The global robustness bound is $\epsilon = 1.5$. REGLO significantly reduces VB, PGD-B, and PGD-R compared with ST. Moreover, only ST+REGLO guarantees satisfaction of the global robustness property (VB $< \epsilon$). For the three baseline methods, their VBs all exceed the required bound as the non-zero PGD-Rs also indicate the detection of counterexamples. Given that REGLO does not have access to the training data and Auto MPG is a dataset for regression (any deviation of the output will increase the ERR), it is reasonable that the ERR for ST+REGLO is not as good as the rest three methods.

|              |       | ST       | ST+AT-G   | ST+REGLO     |
|--------------|-------|----------|-----------|--------------|
|              | VB    | 5.3125   | 5.219     | **0.906**    |
|              | PGD-B | 0.983    | 0.679     | **0.188**    |
| $X = X_3$    | PGD-R | 94.30%   | 93.87%    | **0.0%**     |
|              | ACC   | **96.35%**| 91.09%   | 96.11%       |
|              | T     | 57.58    | 57.58+6   | 57.58+56.78  |
|              | VB    | 4.726    | 4.453     | **1.004**    |
|              | PGD-B | 0.925    | 0.521     | **0.201**    |
| $X = X_6$    | PGD-R | 89.60%   | 72.69%    | **0.0%**     |
|              | ACC   | **96.35%**| 91.91%   | 96.15%       |
|              | T     | 57.58    | 57.58+6   | 57.58+46.69  |
|              | VB    | 4.698    | 4.637     | **0.994**    |
|              | PGD-B | 0.875    | 0.636     | **0.181**    |
| $X = X_9$    | PGD-R | 87.68%   | 66.29%    | **0.0%**     |
|              | ACC   | **96.35%**| 91.27%   | 96.27%       |
|              | T(s)  | 57.58    | 57.58+6   | 57.58+47.85  |

Table 3: Comparing ST + REGLO with ST and ST+AT-G on MNIST. REGLO significantly reduces VB, PGD-B, and PGD-R with negligible accuracy drops. In comparison, fine-tuning (using AT-G) does not provide much reduction to any of the three global robustness-related metrics and results in a much larger drop in accuracy.

We consider $X$ to be a class rectangle (the hyper-rectangle that contains all the training inputs with the same class label). For example, $X_k = \{x \mid x^{l_k} \leq x \leq x^{u_k}\}$, where $x_i^{l_k} = \min_{x \in X_k} x_i$ and $x_i^{u_k} = \max_{x \in X_k} x_i$ for every dimension $i$ of the input space and a fixed class $k$.

For REGLO, we search repair areas by random sampling on $X$ and choose 30 areas to repair. A subset of the results is shown in Table 3 and Table 4. The full set of results can be found in the Appendix.

## 5.4 Sound Repair with Stronger Constraints

To address the issue of soundness due to sampling and repairing only a subset of the violating linear regions when the input dimension or the number of linear regions is large, we consider imposing a stronger constraint during repair to recover the guarantee on satisfying the global robustness property. Figure 3 illustrates the results of using a smaller bound $\bar{\epsilon} \leq \epsilon$ for the max constraint in the optimization problem (6). We fix the repair regions to the ones sampled with $\bar{\epsilon} = 0.3$ to eliminate the effect of random sampling for the repair regions (using a smaller $\bar{\epsilon}$ does not affect the overall set of repair regions since the repair regions are still identified using $\epsilon$). At $\bar{\epsilon} = 0.1$, we have a VB less than the required $\epsilon$ of 0.3 and the accuracy drop is less than $5\%$. It is possible to guarantee global robustness over $X$ by repairing only a subset of the violating regions in $X$ because repairs on those regions can generalize to the other regions.

|          |       | AT      | AT+AT-G  | AT+REGLO     |
|----------|-------|---------|----------|--------------|
| $X = X_3$ | VB    | 3.084   | 3.062    | **1.815**    |
|          | PGD-B | 0.429   | 0.386    | **0.249**    |
|          | PGD-R | 4.17%   | 2.0799%  | **0.0%**     |
|          | ACC   | **95.68%** | 95.56% | 95.54%      |
|          | T(s)  | 665     | 665+6    | 665+25.59    |
| $X = X_6$ | VB    | 3.034   | 3.008    | **1.798**    |
|          | PGD-B | 0.355   | 0.396    | **0.277**    |
|          | PGD-R | 1.31%   | 0.51%    | **0.0%**     |
|          | ACC   | **95.68%** | 95.58% | 89.24%      |
|          | T     | 665     | 665+6    | 665+33.64    |
| $X = X_9$ | VB    | 2.981   | 2.972    | **1.583**    |
|          | PGD-B | 0.353   | 0.333    | **0.247**    |
|          | PGD-R | 2.420%  | 0.85%    | **0.0%**     |
|          | ACC   | **95.68%** | 95.55% | 91.19%      |
|          | T     | 665     | 665+6    | 665+39.99    |

Table 4: Comparing AT+REGLO with AT and AT+AT-G on MNIST. Both PGD-B and PGD-R are already very small for a DNN trained with AT and REGLO further reduces VB, PGD-B, and PGD-R with a small accuracy drop. In comparison, fine-tuning (using AT-G) is not able to improve the robustness for the given property especially on VB.

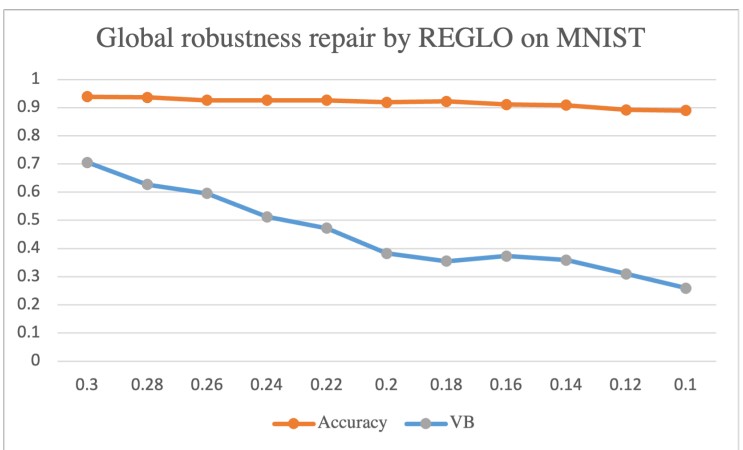

Figure 3: Global robustness repair by REGLO on MNIST with $X = X_0$ (the hyper-rectangle that contains all the training inputs labeled as class 0) and $\epsilon = 0.3$. We use an $\bar{\epsilon} \leq \epsilon$ in place of $\epsilon$ in the optimization problem (6) to impose a stronger constraint. The figure shows that as we decrease $\bar{\epsilon}$ from $0.3$ to $0.1$, we can satisfy the $\epsilon$ bound in the specification with only a small drop in accuracy.

## 6   Concluding Remarks

REGLO is the *first* work that enables provable repair of neural networks for global robustness properties. Experimental results demonstrate the effectiveness of the approach against multiple baselines. Achieving such deterministic guarantees, however, can be challenging when we apply the technique to larger networks. Compared to local robustness, global robustness is a strictly stronger condition that requires robustness for all infinitely many inputs within a region. As a result, verifying whether a network satisfies a given global robustness property is fundamentally more challenging than local robustness verification. ITNE [20], which REGLO uses as a subroutine as necessitated by the need for deterministic guarantees, is the current state-of-the-art technique for verifying global robustness properties. However, it can still take hours for ITNE to compute a global robustness bound for a network with around 10k neurons on the CIFAR-10 dataset. Another challenge to scalability lies in finding the violating linear regions. As the size of the network increases, solving the MILP problem in Eq. (3) can become very expensive. In addition, the number of linear regions of a ReLU network grows exponentially in the number of layers and polynomially in the number of neurons (or layer width) [52]. Thus, while in theory REGLO can provide completeness guarantees as stated in Theorem 2,

for large ReLU DNNs (and non-CPWL DNNs) we have to resort to random sampling for finding the violating regions as described in Section 3.2. Future works include improving the efficiencies of identifying repair areas and computing tight verified global robustness bounds. Another direction is to consider statistical techniques to sidestep the inherent complexity of aiming for deterministic guarantees for global robustness properties.

**Acknowledgements.** We gratefully acknowledge the support from the National Science Foundation awards CCF-1646497, CCF-1834324, CNS-1834701, CNS-1839511, IIS-1724341, CNS-2038853, ONR grant N00014-19-1-2496, the Intelligence Advanced Research Projects Agency (IARPA) under the contract W911NF20C0038, and the US Air Force Research Laboratory (AFRL) under contract number FA8650-16-C-2642.

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

# 7 Appendix

## 7.1 Proofs of Theorems

**Corollary 1** (Gradient Features of Global Robustness Properties). *For a DNN $f$ and a global robustness property $\mathbf{P}(X, \Omega, \epsilon)$, if there is a counterexample $(x, \Delta x)$ such that $\Delta x \in \Omega$ and $\|f(x + \Delta x) - f(x)\| \geq \epsilon$, then there exists a differentiable point $x'$ between $x$ and $x + \Delta x$ such that $\|x - x'\| \leq \frac{dia(\Omega)}{2}$ and $\|\frac{\partial f}{\partial x}(x')\| > \frac{\epsilon}{dia(\Omega)}$, where $dia(\Omega)$ is the diameter of $\Omega$.*

*Proof.* Consider $g : [0, 1] \to \mathbb{R}^n$ defined as $g(t) = f(x + t\Delta x)$ and $g$ is piecewise differentiable on $[0, 1]$. Since $\epsilon < \|g(1) - g(0)\|$, there must exist a differentiable interval $(t_i, t_{i+1})$, such that

$$(t_{i+1} - t_i)\epsilon < \|g(t_{i+1}) - g(t_i)\|$$

Then by Theorem 1, we have,

$$
\begin{aligned}
(t_{i+1} - t_i)\epsilon &< \|g(t_{i+1}) - g(t_i)\| \\
&\leq (t_{i+1} - t_i) \sup_{t \in (t_i, t_{i+1})} \|g'(t)\| \\
&= (t_{i+1} - t_i) \sup_{t \in (0,1)} \|\frac{\partial f}{\partial x}(x + t\Delta x)\Delta x\|
\end{aligned}
$$

Thus, there exists a $x' = x + t'\Delta x$, $\|x - x'\| \leq \frac{dia(\Omega)}{2}$ (we can simply switch $x$ and $x + \Delta x$ if $\|x - x'\| > \frac{dia(\Omega)}{2}$), and

$$\|\frac{\partial f}{\partial x}(x')\Delta x\| > \epsilon \Rightarrow \|\frac{\partial f}{\partial x}(x')\| > \frac{\epsilon}{dia(\Omega)}$$

$\square$

**Property 1.** *Optimization problem (6) is convex and thus any local minimum also achieves the global minimum.*

*Proof.* By the definition of a norm, we have any norm $\|.\|$ is a convex function.

For any $\Delta\theta_1$, $\Delta\theta_2$, $\lambda \in [0, 1]$ and any $\Delta z_i$, we have

$$
\begin{aligned}
&\max_{[\Delta x, \Delta z_i]} \|(\theta + \lambda\Delta\theta_1 + (1 - \lambda)\Delta\theta_2)\Delta z_i\| \\
&= \max_{[\Delta x, \Delta z_i]} \|\lambda(\theta + \Delta\theta_1)\Delta z_i + (1 - \lambda)(\theta + \Delta\theta_2)\Delta z_i\| \\
&\leq \max_{[\Delta x, \Delta z_i]} [\|\lambda(\theta + \Delta\theta_1)\Delta z_i\| + \|(1 - \lambda)(\theta + \Delta\theta_2)\Delta z_i\|] \\
&\leq \max_{[\Delta x, \Delta z_i]} \|\lambda(\theta + \Delta\theta_1)\Delta z_i\| \\
&+ \max_{[\Delta x, \Delta z_i]} \|(1 - \lambda)(\theta + \Delta\theta_2)\Delta z_i\| \\
&\leq \lambda\epsilon + (1 - \lambda)\epsilon = \epsilon
\end{aligned}
$$

Both the objective and the constraints are convex. Therefore, we have that optimization programming (6) is convex. Thus, any local minimum achieves the global minimum. $\square$

**Theorem 2** (Completeness Guarantees). *We have the following completeness guarantees for* `REGLO`*:*

1. *For a Single Iteration Repair,* `REGLO` *can always find a solution to the optimization problem (6).*

2. *For an Iterative Repair on a piecewise linear DNN,* `REGLO` *always terminates with no more repair areas to be found by solving MILP (3).*

*Proof.* For a *Single Iteration Repair*, the optimization problem (6) is convex over a close domain. Since $\Delta\theta = -\theta$ is a feasible solution, the domain where $\Delta\theta$ is in is always feasible. Therefore, optimization problem (6) has an optimal solution. Since the optimization problem is convex, the optimal solution is unique and `REGLO` can always find the optimal solution to the optimization problem (6).

For an *Iterative Repair* on a CPWL DNN, `REGLO` can always find an optimal solution for every *Single Iteration Repair*. Given that the number of linear regions for a CPWL DNN is finite, `REGLO` always terminates with no more repair areas to be found by solving the MILP. $\square$

**Theorem 3** (Limited Side Effect for Single Iteration Repair). *Given a global robustness property* $\mathbf{P}(X, \Omega, \epsilon)$*, a target DNN* $f$*, and weight change* $\Delta\theta$ *from a Single Iteration Repair, we have*

  1. *for any area* $\mathcal{B} \subset \cup_{i \in I} \mathcal{A}_i \subset X$, $\widehat{f} \models \mathbf{P}(\mathcal{B}, \Omega, \epsilon)$;

  2. *for any area* $\mathcal{C} \subset X$ *which is not a subset of* $\cup_{i \in I} \mathcal{A}_i$, $\widehat{f} \models \mathbf{P}(\mathcal{C}, \Omega, \epsilon + 2L\|\Delta\theta\|\|X\|)$, *where* $L$ *is the Lipschitz constant of* $f$ *from the input layer to the last hidden layer.*

*Proof.* The constraints of optimization problem (6) ask the resulting DNN must satisfy the global robustness property on $\mathcal{B} \subset \cup_{i \in I} \mathcal{A}_i \subset X$, a subset of the repair areas.

The difference between the resulting DNN $\widehat{f}$ and the target DNN $f$ on any input $x$ can be bounded by the norm of $\Delta\theta$: $\|\widehat{f}(x) - f(x)\| = \|(\theta + \Delta\theta)f_{n-1}(x) - \theta f_{n-1}(x)\| = \|\Delta\theta f_{n-1}(x)\| \leq \|\Delta\theta\| \cdot \|f_{n-1}\| \cdot \|x\| = L\|\Delta\theta\| \cdot \|x\|$, where $f_{n-1}$ is the DNN function from the input layer to the last hidden layer.

For any $x \in \mathcal{C} \subset X$, which is not in a subset of $\cup_{i \in I} \mathcal{A}_i$, and any $\Delta x \in \Omega$:

$$
\begin{aligned}
&\|\widehat{f}(x + \Delta x) - \widehat{f}(x)\| \\
=&\|\widehat{f}(x + \Delta x) - f(x + \Delta x) \\
&+ f(x + \Delta x) - f(x) + f(x) - \widehat{f}(x)\| \\
\leq&\|\widehat{f}(x + \Delta x) - f(x + \Delta x)\| + \|f(x + \Delta x) - f(x)\| \\
&+ \|f(x) - \widehat{f}(x)\| \\
\leq& L\|\Delta\theta\| \cdot \|x + \Delta x\| + \epsilon + L\|\Delta\theta\| \cdot \|x\| \\
\leq& \epsilon + 2L\|\Delta\theta\|\|X\|
\end{aligned}
$$

$\square$

**Corollary 2** (Soundness Guarantees for Repairing CPWL DNNs). *Given a global robustness property* $\mathbf{P}(X, \Omega, \epsilon)$*, a piecewise linear DNN* $f$*, and weight change* $\Delta\theta$ *from Iterative Repair,* `REGLO` *will terminate with no more repair areas to be found and the resulting DNN* $\widehat{f} \models \mathbf{P}(X, \Omega, \epsilon)$.

*Proof.* For *Iterative Repair* on a CPWL DNN $f$, by Theorem 2, `REGLO` will terminate with no more repair areas to be found. Therefore, the resulting DNN satisfies $\mathbf{P}(X, \Omega, \epsilon)$ outside the repair areas. In addition, by Theorem 3, we have that the resulting DNN satisfies $\mathbf{P}(\mathcal{C}, \Omega, \epsilon)$ for any $\mathcal{C} \subset X$. Combining these results, we have that `REGLO` will terminate with no more repair areas to be found and the resulting DNN $\widehat{f} \models \mathbf{P}(X, \Omega, \epsilon)$. $\square$

**Corollary 3** (Soundness Guarantee for Repairing General DNNs). *Given a global robustness property* $\mathbf{P}(X, \Omega, \epsilon)$*, a DNN* $f$*, and weight change* $\Delta\theta$ *from Iterative Repair,* `REGLO` *returns a DNN* $\widehat{f} \models \mathbf{P}(\mathcal{C}, \Omega, \epsilon + 2L\|\Delta\theta\|\|X\|)$ *for any repair area* $\mathcal{C}$.

*Proof.* Since *Iterative Repair* collects both the previously repaired areas and the new repair areas found at one iteration, the last *Single Iteration Repair* will repair all those collected repair areas. Therefore, by Theorem 3, the last *Single Iteration Repair* will return a DNN $\widehat{f}$ that satisfies $\mathbf{P}(\mathcal{C}, \Omega, \epsilon + 2L\|\Delta\theta\|\|X\|)$. $\square$

**Algorithm 2** Repair via Barrier Method

---

**Input**: Current last-layer weight $\theta$, repair areas $\{\mathcal{A}_i\}_{i \in I}$, and $\Omega$.
**Parameter**: Initial step size $\alpha$, initial weight of barrier function $t$, an early stop threshold $\delta$, and maximal steps $K$.
**Output**: $\Delta\theta$

1: Let $k = 0$.
2: Let $\Delta\theta = -\theta$ (start from a feasible solution).
3: **while** $k < K$ **do**
4:     Let $\epsilon_i^*$ and $\Delta z_i^*$ be the optimal value and optimal solution of optimization problem (5), respectively.
5:     Compute the gradient $g = \frac{\partial \epsilon_i^*(\theta + \Delta\theta)}{\partial \Delta\theta}$ according to Equation (8).
6:     **if** $\|g\| < \delta$ **then**
7:         Break
8:     **end if**
9:     Update $\Delta\theta = \Delta\theta - \alpha \cdot g$
10:     Update $\alpha$ and $t$.
11: **end while**
12: **return** $\Delta\theta$.

---

## 8 Additional Experiment Details

### 8.1 DNN Architectures and Training/Repair Hyperparameters

German Credit: The DNN is a multi-layer perceptron with ReLU activation functions. It has an input layer with 20 neurons, 2 hidden layers with 200 neurons in each layer, and a final output layer with 1 neuron.

Auto MPG: The DNN is a multi-layer perceptron with ReLU activation functions. It has an input layer with 3 neurons, 2 hidden layers with 20 neurons in each layer, and a final output layer with 1 neuron.

MNIST: The DNN is a convolutional neural network with ReLU activation functions. It has two convolutional layers and one dense layer. The two convolutional layers have 16 and 32 channels, respectively. The dense layer has 16 neurons.

We set the learning rate to $10^{-3}$ in all the training experiments. For fine-tuning, we set a learning rate of $5 * 10^{-4}$ on the German Credit experiment, $2 * 10^{-4}$ on the Auto MPG experiment, and $2 * 10^{-4}$ on the MNIST experiment. The number of epochs for training is 20 and the batch size is 256.

For AT-G, the number of points that we sampled in each iteration is 100.

The step size we used for PGD-B and PGD-R is $0.01$ and the number of steps is 200. The step size we used for AT and AT-G is $0.01$ and the number of steps is 10.

### 8.2 Full Results for Norm-bounded Global Robustness Repair on MNIST

Table 5 shows the full set of results for norm-bounded global robustness repair on MNIST. Overall, ST+REGLO achieves the best (smallest) VB and AT-G achieves the best (smallest) PGD-B, while both methods have very small PGD-R (0.0% or close to 0.0%) and do not have any significant drop in accuracy. Compared with AT-G, ST+REGLO has the following advantages: (1) repair using REGLO does not require access to the training data (which may not be available due to privacy reasons for instance ), (2) it can be applied to a trained network as a post-hoc modification especially when the global robustness property is only given after training, and (3) a much smaller runtime as REGLO does not require adversarial training. In addition, AT-G is not able to reduce VB which is required by the specification. On the other hand, we can use a $\bar{\epsilon} < \epsilon$ in REGLO to satisfy the specification even if we are only sampling and repairing a subset of the violating linear regions, as shown at the beginning of the Appendix.

|  |  | ST | AT | AT-G | ST+AT-G | AT+AT-G | ST+REGLO | AT+REGLO |
|---|---|---|---|---|---|---|---|---|
| $X = X_0$ | VB | 5.577 | 3.08 | 4.459 | 5.237 | 3.0472 | 0.7053 | 1.608 |
|  | PGD-B | 0.9367 | 0.3641 | 0.0701 | 0.6179 | 0.3786 | 0.1764 | 0.2514 |
|  | PGD-R | 94.30% | 3.37% | 0.0% | 78.779% | 1.659% | 0.0% | 0.0% |
|  | ACC | 96.35% | 95.68% | 96.38% | 91.1% | 95.57% | 93.86% | 89.86% |
|  | T | 57.58 | 665 | 1171 | 57.58+6 | 665+6 | 57.58+38.83 | 665+33.99 |
| $X = X_1$ | VB | 5.548 | 3.019 | 5.799 | 4.379 | 3.183 | 0.807 | 1.831 |
|  | PGD-B | 0.9279 | 0.360 | 0.0686 | 0.6243 | 0.3569 | 0.1538 | 0.3057 |
|  | PGD-R | 93.73% | 2.84% | 0.0% | 92.11% | 1.439% | 0.0% | 0.030% |
|  | ACC | 96.35% | 95.68% | 96.81% | 91.36% | 95.57% | 93.79% | 86.20% |
|  | T | 57.58 | 665 | 1165 | 57.58+6 | 665+6 | 57.58+48.98 | 665+35.02 |
| $X = X_2$ | VB | 4.860 | 3.2468 | 4.515 | 4.479 | 3.063 | 0.8838 | 1.963 |
|  | PGD-B | 1.0158 | 0.4111 | 0.0907 | 0.6516 | 0.3912 | 0.1815 | 0.3057 |
|  | PGD-R | 95.92% | 4.36% | 0.0% | 90.79% | 2.410% | 0.0% | 0.02% |
|  | ACC | 96.35% | 95.68% | 95.96% | 91.49% | 95.54% | 96.17% | 86.23% |
|  | T | 57.58 | 665 | 1188 | 57.58+6 | 665+6 | 57.58+45.98 | 665+39.08 |
| $X = X_3$ | VB | 5.3125 | 3.084 | 5.159 | 5.219 | 3.0624 | 0.9064 | 1.815 |
|  | PGD-B | 0.9832 | 0.4295 | 0.0962 | 0.6791 | 0.3867 | 0.1881 | 0.2497 |
|  | PGD-R | 94.30% | 4.17% | 0.0% | 93.87% | 2.0799% | 0.0% | 0.0% |
|  | ACC | 96.35% | 95.68% | 95.97% | 91.09% | 95.56% | 96.11% | 95.54% |
|  | T | 57.58 | 665 | 1174 | 57.58+6 | 665+6 | 57.58+56.78 | 665+25.59 |
| $X = X_4$ | VB | 5.143 | 3.277 | 5.6209 | 4.363 | 3.188 | 0.8457 | 1.689 |
|  | PGD-B | 0.9692 | 0.3763 | 0.0980 | 0.6818 | 0.3962 | 0.1722 | 0.2206 |
|  | PGD-R | 94.61% | 4.41% | 0.0% | 87.23% | 2.25% | 0.0% | 0.0% |
|  | ACC | 96.35% | 95.68% | 96.27% | 91.39% | 95.55% | 96.03% | 95.09% |
|  | T | 57.58 | 665 | 1097 | 57.58+6 | 665+6 | 57.58+53.81 | 665+29.05 |
| $X = X_5$ | VB | 4.786 | 3.094 | 5.627 | 4.748 | 3.1461 | 0.8528 | 1.747 |
|  | PGD-B | 1.0001 | 0.3866 | 0.0754 | 0.6771 | 0.3680 | 0.1708 | 0.3285 |
|  | PGD-R | 93.56% | 5.11% | 0.0% | 81.72% | 2.370% | 0.0% | 1.46% |
|  | ACC | 96.35% | 95.68% | 96.51% | 91.00% | 95.54% | 95.98% | 86.2% |
|  | T | 57.58 | 665 | 1143 | 57.58+6 | 665+6 | 57.58+39.47 | 665+26.70 |
| $X = X_6$ | VB | 4.726 | 3.034 | 4.767 | 4.453 | 3.008 | 1.004 | 1.798 |
|  | PGD-B | 0.9255 | 0.3557 | 0.0561 | 0.5215 | 0.3964 | 0.2011 | 0.2772 |
|  | PGD-R | 89.60% | 1.31% | 0.0% | 72.69% | 0.519% | 0.0% | 0.0% |
|  | ACC | 96.35% | 95.68% | 96.5% | 91.91% | 95.58% | 96.15% | 89.24% |
|  | T | 57.58 | 665 | 1157 | 57.58+6 | 665+6 | 57.58+46.69 | 665+33.64 |
| $X = X_7$ | VB | 4.668 | 3.033 | 4.787 | 4.810 | 3.0184 | 0.9147 | 1.715 |
|  | PGD-B | 0.8889 | 0.3690 | 0.0962 | 0.6691 | 0.3672 | 0.1695 | 0.3025 |
|  | PGD-R | 85.90% | 2.44% | 0.0% | 82.23% | 1.070% | 0.0% | 0.01% |
|  | ACC | 96.35% | 95.68% | 95.93% | 91.18% | 95.53% | 96.21% | 86.21% |
|  | T | 57.58 | 665 | 1184 | 57.58+6 | 665+6 | 57.58+54.13 | 665+27.74 |
| $X = X_8$ | VB | 4.728 | 3.030 | 5.875 | 5.127 | 3.078 | 1.032 | 1.741 |
|  | PGD-B | 0.927 | 0.3933 | 0.1111 | 0.6179 | 0.3441 | 0.2057 | 0.2976 |
|  | PGD-R | 91.47% | 4.40% | 0.0% | 75.12% | 1.920% | 0.0% | 0.0% |
|  | ACC | 96.35% | 95.68% | 96.58% | 90.87% | 95.52% | 96.16% | 86.2% |
|  | T | 57.58 | 665 | 1155 | 57.58+6 | 665+6 | 57.58+42.92 | 665+30.79 |
| $X = X_9$ | VB | 4.698 | 2.981 | 5.797 | 4.637 | 2.972 | 0.9949 | 1.583 |
|  | PGD-B | 0.875 | 0.3531 | 0.0656 | 0.6362 | 0.3333 | 0.1810 | 0.2478 |
|  | PGD-R | 87.68% | 2.420% | 0.0% | 66.29% | 0.85% | 0.0% | 0.0% |
|  | ACC | 96.35% | 95.68% | 96.96% | 91.27% | 95.55% | 96.27% | 91.19% |
|  | T | 57.58 | 665 | 1179 | 57.58+6 | 665+6 | 57.58+47.85 | 665+39.99 |

Table 5: Global robustness repair on MNIST for different class rectangles.

