# OpenReview forum: "REGLO: Provable Neural Network Repair for Global Robustness Properties"
_NeurIPS.cc/2022/Workshop/TSRML — TSRML2022_

### Official Review · Reviewer_PhKs · 2022-10-12

**Overall Recommendation:** Overall this is a great paper.
**Overall Rating:** 8

**Summary:**

This paper proposes to provably repair DNN for global robustness. MILP is used to find violation regions. Fixes are performed by updating the weights on the last layer with an optimization.

**Strengths:**

* The method is claimed to be the first technique on provable DNN repair for global robustness.
* The problem is important and the method is sound and well motivated.
* Experiments are solid.


**Weaknesses:**

* So far the scalability seems to be limited.


**Review Confidence:**

3: The reviewer is fairly confident that the evaluation is correct

---

### Official Review · Reviewer_Veie · 2022-10-17
**A well-written paper with significant contributions towards repair of neural networks for global properties**

**Overall Rating:** 7

**Summary:**

The paper proposes REGLO, an approach for the repair of global properties in neural networks. REGLO utilises sound robustness bounds to formulate an optimisation problem for repairing violations with minimal weight changes to the weights in the network's last hidden layer. The paper proves that REGLO is sound and complete.


**Strengths:**

1. The paper is well-written and easy to follow.

2. The topic of the paper is of high interest to the research community.

3. Claims made in the paper are supported by a detailed theoretical and empirical analysis.

**Weaknesses:**

While the advantages of the REGLO algorithm are well supported by theoretical and empirical results, the discussion on REGLO's limitations is somewhat lacking. In particular, addressing the following two issues would significantly strengthen the paper.

1. It seems to me that one of the main limitations of the approach presented in this paper may be its scalability to larger networks. However, the main body of the paper does not seem to include a discussion on scalability, nor does it describe the architectures of the networks used in the empirical evaluation. Some of this information is provided in the supplementary material; however, for the MNIST network, not enough details are given to calculate the number of ReLU activations which may be essential for scalability. In my opinion, a detailed discussion of this topic in the main body of the paper would significantly strengthen the paper.

2. The optimisation problem in Equation (6) aims to find minimal weight changes in order to preserve the network's performance (e.g. accuracy). However, earlier work in repair has pointed out that minimal weight changes are often not a good heuristic for preserving performance (See e.g. Repairing misclassifications in neural networks using limited data (https://dl.acm.org/doi/abs/10.1145/3477314.3507059)). In my opinion, this issue warrants a more detailed discussion.

**Overall Recommendation:**

The paper has significant contributions that are of substantial interest to the field. However, the discussion on limitations of the proposed method is somewhat lacking; expanding this discussion would significantly strengthen the paper.

**Review Confidence:**

3: The reviewer is fairly confident that the evaluation is correct

---

### Official Review · Reviewer_Vb8N · 2022-10-21
**Good paper**

**Overall Rating:** 7

**Summary:**

The paper presents a technique for repairing DNNs in order to satisfy some global robustness properties. The key idea is to leverage the fact that a datapoint violating the global robustness property has a large gradient. The paper proposes a minimal weight change in the last hidden layer of the DNN to fix this violation. The paper guarantees for piecewise linear DNNs, a repair is guaranteed to be found and the resulting model is guaranteed to satisfy the given global robustness property. The paper conducts extensive experiments and compares various baselines to establish the effectiveness of their method across a number of benchmarks.

**Strengths:**

- The paper is sound and the DNN repair problem is well motivated.
- The illustrative diagrams provided in the paper makes it easier to gain intuition into and follow the paper, and also to better understand the proposed method, especially, Figures 1 and 2.
- The experiments are thorough and the authors compare their results to a wide range of baselines, showing the superiority of the presented method.


**Weaknesses:**

- I noticed the experiments are done on only small datasets (e.g. MNIST). What is the main bottleneck from testing the proposed method on large-scale datasets/models (e.g. ImageNet)? Is it the model size? The input dimension? The limitations of verification methods for large models? It would be good if the authors can comment on this in the paper and discuss the limitations of the proposed method.


Minor:
- Many of the inline citations need fixing (should be in brackets/parentheses)


**Overall Recommendation:**

Overall, I think this is a good paper that deserves being presented in the workshop. I recommend acceptance.

**Review Confidence:**

3: The reviewer is fairly confident that the evaluation is correct

---

### Decision · Program_Chairs · 2022-10-23

Accept